# Caffeine and Chlorogenic Acid Combination Attenuate Early-Stage Chemically Induced Colon Carcinogenesis in Mice: Involvement of oncomiR miR-21a-5p

**DOI:** 10.3390/ijms23116292

**Published:** 2022-06-04

**Authors:** Ariane Rocha Bartolomeu, Guilherme Ribeiro Romualdo, Carmen Griñán Lisón, Zein Mersini Besharat, Juan Antonio Marchal Corrales, Maria Ángel García Chaves, Luís Fernando Barbisan

**Affiliations:** 1Department of Pathology, Botucatu Medical School, São Paulo State University (UNESP), Botucatu 18618-687, SP, Brazil; bartolomeuar@gmail.com (A.R.B.); guilherme.romualdo@unesp.br (G.R.R.); 2Department of Structural and Functional Biology, Biosciences Institute, São Paulo State University (UNESP), Botucatu 18618-689, SP, Brazil; 3Biosanitary Research Institute of Granada (ibs.GRANADA), University Hospitals of Granada, University of Granada, 18012 Granada, Spain; carmen.grinan@genyo.es (C.G.L.); jmarchal@ugr.es (J.A.M.C.); mangelgarcia@ugr.es (M.Á.G.C.); 4GENYO (Centre for Genomics and Oncological Research), Pfizer/University of Granada/Andalusian Regional Government, 18016 Granada, Spain; 5UGC de Oncología Médica, Complejo Hospitalario de Jaen, 23007 Jaen, Spain; 6Department of Experimental Medicine, Sapienza University of Rome, 00185 Rome, Italy; zeinmersini.besharat@uniroma1.it; 7Department of Human Anatomy and Embryology, School of Medicine, University of Granada, 18016 Granada, Spain; 8Department of Biochemistry and Molecular Biology III and Immunology, University of Granada, 18016 Granada, Spain

**Keywords:** caffeine, chlorogenic acid, chemically induced colon carcinogenesis, miRNA expression, mice

## Abstract

Colorectal cancer (CRC) is one of most common cancers worldwide, with high rates of mortality. Epidemiological findings demonstrate that coffee consumption reduces the risk of developing CRC by ~13%. In general, in vivo and in vitro findings demonstrate the antiproliferative, antioxidant and proapoptotic effects of brewed coffee or major bioavailable coffee compounds. Thus, it was assessed whether caffeine (CAF) and/or chlorogenic acid (CGA) attenuates the early-stage of chemically induced mouse colon carcinogenesis. Male Swiss mice were submitted to a 1,2-dimethylhydrazine/deoxycholic acid (DMH/DCA)-induced colon carcinogenesis model. These animals received CAF (50 mg/kg), CGA (25 mg/kg) or CAF+CGA (50 + 25 mg/kg) intragastrically for five times/week for ten weeks. CAF+CGA had the most pronounced effects on decreasing epithelial cell proliferation (Ki-67) and increasing apoptosis (cleaved caspase-3) in colonic crypts. This treatment also decreased the levels of proinflammatory cytokines IL-6, IL-17 and TNF-α, and downregulated the oncomiR miR-21a-5p in the colon. Accordingly, the analysis of miR-21a-5p targets demonstrated the genes involved in the negative regulation of proliferation and inflammation, and the positive regulation of apoptosis. Ultimately, CAF+CGA attenuated preneoplastic aberrant crypt foci (ACF) development. Our findings suggest that a combination of coffee compounds reduces early-stage colon carcinogenesis by the modulation of miR-21a-5p expression, highlighting the importance of coffee intake to prevent CRC.

## 1. Introduction

Colorectal cancer (CRC) is the third most common cancer and ranks second in worldwide mortality. More than 1.9 million new cases and 935 thousand deaths were estimated in 2020 [1,2]. In addition, cases/deaths are expected to rise by ~60% within the next 15 years [1,2]. Most CRC cases are classified as sporadic (55–85%), and more than half are related to environmental factors, such as the high consumption of processed red meat, alcoholism, smoking, a sedentary lifestyle and obesity [3,4,5]. The natural development of CRC is considered a multistep process that occurs in four main stages (initiation, promotion, progression and metastasis), that includes the emergence and accumulation of molecular and morphological alterations [6]. Most CRC cases arise from an adenoma–carcinoma multistep sequence, involving multiple epigenetic and genetic alterations, including aberrant microRNA (miRNA) expression [7]. Deregulated miRNA expression has been gaining relevance over the past few decades [8,9]. miRNAs are small non-coding RNAs that regulates gene expression by mRNA degradation or translational repression. These small molecules may act as oncogenes or tumor suppressors, and their differential expression is considered a hallmark of colon carcinogenesis [9,10]. While silencing or triggering different signaling pathways, the miRNAs may contribute to the transition from normal colonocytes to neoplastic stages. In particular, the miR-21a-5p is upregulated in both adenomas and adenocarcinomas, and the expression of this miRNA is positively correlated to disease staging, indicating a potential role of miR-21a-5p in the CRC development [10]. This oncomiR negatively regulates the tumor suppressor PDCD4 at the posttranscriptional level, via a specific target site within the 3′-UTR inducing invasion and metastasis in several CRC cell lines [10,11]. Moreover, miR-21 can inhibit apoptosis by altering anti-apoptotic Bcl-2, and gemcitabine-induced apoptosis is specifically inhibited by miR-21 via PTEN and PI3K pathway [11,12].

In order to understand the different aspects of colon tumorigenesis, chemically induced murine models were established, including those induced by 1,2-dimethylhidrazine (DMH) and its metabolite, azoxymethane (AOM) [13], especially when associated with promoting a stimulus using bile acids such as deoxycholic acid (DCA) [14]. These bioassays provided information on the morphological and molecular features comparable to the corresponding human diseases, enabling a translational screening of preventive and therapeutic strategies towards this malignancy [13,15]. Single or multiple doses of DMH/AOM induce preneoplastic lesions and tumors mainly in the middle and distal colon. Aberrant crypt foci (ACF), constituting of one or more aberrant crypts (AC), are considered the earliest endpoint lesions during colorectal carcinogenesis, classified as putative precursors for human and rodent CRC [16,17,18]. Thus, ACF detection and quantification have been proposed to identify potential preventive agents in chemically induced CRC bioassays [13,15].

Epidemiological and experimental findings strongly suggest that nutritional habits and lifestyle display key roles in CRC development, and interventions in these risk factors should be the primary prevention for this cancer [4,19]. Coffee consumption is based on the intake of brewed and espresso coffee beverages, prepared with beans that have been roasted and ground, originally from the *Coffea* genus plant species [20,21,22]. The higher annual rate of consumption per capita is concentrated in the northern hemisphere, especially in European countries (6.29–9.12 kg per capita), followed by the US (~4 kg per capita) then Asian and African countries (~3 kg per capita) [20,21,22]. Drinking coffee is generally considered a healthy habit [2]. Although evidence regarding the influence of drinking coffee on the risk of CRC is limited, some epidemiological studies clearly demonstrate a positive correlation between coffee consumption and a reduced risk of CRC or an improvement in the overall survival rate from cancer; however, other studies demonstrated no correlation in stimulating novel investigations [22,23,24]. Indeed, the protection or a reduction in the risk of the upper and lower digestive tract malignancies are associated with a mild and high intake of coffee beverages (≥3 cups a day) [22]. Guercio et al. (2015) demonstrated a significant decrease in the recurrence of cancer and death in patients with stage III CRC who consumed four cups/day [25]. Recently, a prospective study involving 1171 patients with advanced or metastatic CRC demonstrated that a high coffee consumption (≥3 cups a day) improved the overall survival (OS) rates, reducing the therapeutic side effects and risk of cancer progression [26].

Coffee is considered a complex pharmacopeia, and the composition of its extract depends on many factors that vary from harvesting the seeds to serving it [27]. In terms of abundance, the main bioactive compound present in the beverage coffee is caffeine (CAF, 1,3,7-trimethylxanthine), followed by conjugated hydroxycinnamates commonly referred to as chlorogenic acid (CGA, caffeoylquinic acid). Other bioactive compounds in coffee beverages comprise amino acids, polysaccharides, and melanoidins [22,27,28]. Most of the literature is focused on the beneficial effects, as a whole, of coffee beverages or coffee compounds. Individual and mechanistical studies on the combination of the most abundant compounds against CRC are warranted. Given that coffee bioavailable compounds seem to modulate miRNA biosynthesis and downstream targets [29], it would be important to unveil whether coffee compounds could modify colon carcinogenesis by altering miRNA expression.

Thus, we assessed whether the administration of two of the main bioavailable coffee compounds (individually or combined) attenuates early-stage chemically induced colon carcinogenesis using a DMH/DCA mouse model. In addition, the global miRNA profile was evaluated in order to investigate the involvement of these molecules in the potential chemo preventive properties of coffee compounds. This is the first translational investigation regarding the combination of the two main bioactive compounds (CGA + CAF) against colon carcinogenesis and may provide novel mechanistic insights using doses that resemble human coffee consumption.

## 2. Results

### 2.1. General Findings during Bioactive Coffee Compounds Interventions

Firstly, the effect of coffee compounds on food consumption were analyzed. Body and liver weights from mice, submitted to the DMH/DCA-induced colon carcinogenesis, were taken. The animals exhibited similar initial body weights (Table 1). The DMH/DCA protocol and different interventions with bioactive coffee compounds did not alter food consumption, body weight gain or final body weight. Although interventions did not modify absolute liver weight, all groups submitted to the DMH/DCA protocol demonstrated a significant increase in relative liver weight (%) at the end of week 13 (*p* < 0.05), and coffee compounds did not modify this DMH/DCA-induced effect (Table 1).

### 2.2. Analysis of Colonic Preneoplastic AC and ACF

After a whole-mount colonic mucosa analysis, all groups that received coffee compound interventions similarly demonstrated a significant reduction in the mean number of AC and ACF (*p* < 0.0001 and *p* = 0.004, respectively) per area of colon mucosa analyzed when compared to the DMH/DCA counterpart (Figure 1). Representative photomicrographs of ACF stained by methylene blue, as shown in Figure 1.

### 2.3. Proliferation and Apoptosis Indexes in Colonic Crypts

After a colonic epithelium analysis, all groups submitted to the DMH/DCA-induced mouse model demonstrated an increase in cell proliferation in colonocytes compared with the untreated counterpart (*p* < 0.0001) measured by immunoreactivity for Ki-67. CAF or CGA interventions individually did not change this DMH/DCA-mediated effect on colonocyte proliferation. However, only the CAF+CGA intervention significantly reduced colonocyte proliferation in normal-appearing crypts compared to the DMH/DCA counterpart (*p* < 0.0001) (Figure 2). Moreover, only the CAF+CGA intervention significantly increased (*p* < 0.0001) the percentage of apoptotic colonocytes in the crypts, measured by immunoreactivity for cleaved-caspase 3 (Figure 2).

### 2.4. Proinflammatory Cytokines Analysis

The histological analysis showed that the DMH/DCA group presented more inflammatory infiltrate of lymphocytes, plasma cells and neutrophils in the lamina propria compared to the untreated counterpart. In addition, this colonic low-grade inflammation induced by DCA treatment was reduced by treatments with isolated or combined bioactive coffee compounds in comparison with the DMH/DCA counterpart (Figure 3). In concordance with histological findings, the colonic cytokines analysis demonstrated that the DMH/DCA group notably increased colonic levels of IL-6 (*p* < 0.0001), IL-17 (*p* < 0.0001) and TNF-α (*p* = 0.0003) compared to the untreated counterpart, showing the activation of pro-inflammatory axis (Figure 3). In contrast, treatments with isolated or combined bioactive coffee compounds similarly reduced the expression of IL-6, IL-17 and TNF-α in comparison with the DMH/DCA counterpart (Figure 3).

### 2.5. Global miRNA Expression

The DMH/DCA model showed the upregulation of six miRNAs and the downregulation of four miRNAs in the colon (Table 2, Figure 4A), including the upregulation of the oncomiR miR-21a-5p, that featured the highest FC compared to the untreated counterparts. As CAF+CGA intervention demonstrated the most pronounced effects on reducing ACF development and colonic cytokine levels (as observed similarly in CAF and CGA treatments), while increasing crypt apoptosis, and decreasing crypt proliferation (exclusive to this group); it was evaluated whether this treatment modulated DMH/DCA-induced effect on colonic miRNA expression. CAF+CGA intervention significantly increased the expression of two miRNAs, while it downregulated four miRNAs compared to the DMH/DCA group (Table 2, Figure 4A). Of those miRNAs differentially expressed in our model, CAF+CGA decreased the expression of the oncomiR miR-21a-5p, as depicted in the Venn diagram (Figure 4A).

### 2.6. Analysis miR-21a-5p Target Genes and Network/Functional Correlation

Considering that miR-21a-5p was downregulated by CAF+CGA treatment, target analysis of this miRNA revealed 35 experimentally validated proteins (Appendix A). Pathway analysis showed that most of these targets are involved with the “negative regulation of ERK1 and ERK2 cascade”, “negative regulation of TGF-β receptor signaling pathway”, and “positive regulation of apoptotic process” (Table 3). In keeping with these findings, Phosphatase and tensin homolog (Pten) and mothers against decapentaplegic homolog 7 (Smad7) were as central nodes in the target network analysis (Figure 4B), as proteins directly involved with the modulated biological processes.

## 3. Discussion

This study aimed at evaluating the beneficial effects of the main coffee alkaloids and polyphenols, CAF and CGA, individually or in association, in a well-established chemically induced colon carcinogenesis bioassay [13,30]. The underlying importance of miRNA modulation was also assessed on the chemopreventive effect of coffee compounds. As elicited in a recent prospective study demonstrating that high coffee consumption (≥3 cups a day) improved the OS of CRC patients [26], further mechanistic studies are needed in order to discriminate which bioactive compounds are involved in this protective effect, and whether these bioactive molecules interact with each other [29]. Our findings may contribute to this gap in the literature, as CAF+CGA intervention had the most pronounced effects on decreasing epithelial cell proliferation (Ki-67) and increasing apoptosis in colonic crypts. This treatment also decreased the levels of proinflammatory cytokines IL-6, IL-17 and TNF-α, and downregulated the oncomiR miR-21a-5p in the colon. Ultimately, CAF+CGA attenuated preneoplastic ACF development. Note that our coffee compound intervention followed a human equivalent dose (HED) translational approach, equivalent to the CAF and ACG contained in three cups of coffee (high coffee consumption).

Previous studies reported that direct contact of colonic mucosa with bioactive coffee compounds, mainly CGA, generate metabolites due to the microbiota metabolization, harboring, in situ, several microbial metabolites in amounts higher or similar to those utilized in the in vitro assays, also reaching extremely low peak plasm concentrations [31,32]. CAF is quickly and practically absorbed entirely in the gastrointestinal tract within 45 min, the smallest part in the upper (~20%) and the largest part (~80%) in the lower gastrointestinal tract, where it is sufficiently hydrophilic to cross biological membranes, reaching the liver, where a biotransformation occurs through the cytochrome P450 (CYP) enzymes, generating metabolites (paraxanthine, dimethylxanthine and theobromine). These metabolites are immediately bioavailable, reaching a plasma peak of ~33 µM within 60–80 min after the higher consumption, equivalent to the intake of three cups/day (~350 mL) of common filtered coffee [33,34]. According to Christopher et al. (2021), the increase in coffee consumption (≥3 cups/day) is inversely proportional to the progression of cancer in patients with advanced or metastatic CRC [35]. Interestingly, this same study observed that when caffeinated and decaffeinated coffee were considered separately, both improved the overall survival; however, the caffeinated coffee presented lower effectiveness on progression-free survival, reinforcing the need for further approaches to identify how these bioactive molecules interact [35].

In our chemically induced mouse model, DMH is a procarcinogen that is biotransformed in the liver into highly reactive ions that alkylate specific genomic DNA bases, resulting in specific DNA adducts, such as O^6^-methylguanine (O^6^-mG) and N^7^-methylguanine (N^7^-mG) in colonic epithelial cells. These DNA adducts can lead to genomic instability and mutation in cancer, a hallmark that significantly contributes to the initiation of rodent colon carcinogenesis [13,36]. In addition to DMH administration, dietary DCA was used to promote the early-stage colon carcinogenesis. Secondary biliary acids are known to contribute to the inflammatory milieu that promotes (pre)neoplastic lesions by different mechanisms, including NLRP3 inflammasome activation that leads to cytokine production [37]. As such, a clear inflammatory colonic context was found in a DMH/DCA model, as IL-6, IL-17 and TNF-α were substantially increased in this group. Although the mechanisms are not fully understood, DCA is also proposed to induce colonic tumors in mice, as a 0.2% DCA in a diet for 8–10 months, without carcinogen initiation, and that led to development of colonic neoplasia [14]. These DMH/DCA-related mechanisms are proposed to induce (pre)neoplastic colon lesions, including ACF, considered putative preneoplastic lesions. The appearance of ACF in both humans and rodents are thought to be the earliest identifiable tumor precursor lesions [16,17,18,30]. The progression of conventional and dysplastic ACF into adenomas and, consequently, into CRC is associated with the accumulation of several genetic and epigenetic changes [15,16]. However, only a small fraction of ACF evolve into an adenoma–carcinoma sequence. Early screening of these lesions is well accepted in short/medium term rodent bioassays as an early marker for CRC prevention [13,15,30]. Indeed, it was found that coffee compounds individually or in combination similarly decreased ACF development. Our findings are in keeping previous bioassays demonstrating that caffeinated coffee [38], CAF [39] or CGA [40] individually attenuated the development of ACF or dysplastic crypts during the initial stages of different chemically induced colon carcinogenesis models in rodents. Nonetheless, the landscape of potential molecular mechanisms involved was not investigated, in detail, in these studies.

It was found that a CAF+CGA combination counteracted the DMH/DCA-induced upregulation of miR-21a-5p, reducing the expression of this oncomiR in the colon. In humans, miR-21-5p is also overexpressed in a colon tumor compared with normal adjacent tissue, and its expression is positively correlated to CRC staging [41,42]. Our DMH/DCA-induced model reflected this marked molecular hallmark of colon carcinogenesis. Increased expression of miR-21-5p in CRC is also associated to a poor prognosis, including poor differentiation, lymph node metastasis and advanced TNM [42]. This miRNA negatively regulates the target gene *Pcdc4*, resulting in increased invasion, migration and cell proliferation of different human CRC cell lines (HT-29, Colo206f, LIM 1863, SW480 and DLD1), contrasting with a knockdown of miR-21-5p cells [43,44]. In addition, it is recognized that tumor suppressor PTEN is inversely associated with miR-21-5p levels in CRC tissues and the HCT-116 colon cancer cell line. When these cells were transfected with miR-21-5p inhibitor, proliferation and migration were suppressed while PTEN protein levels were increased [39]. These findings elicit that miR-21-5p targets tumor suppressor PTEN at the post-transcriptional level, attenuating the PTEN/PI3K/Akt signaling pathway, which is involved in the negative regulation of proliferation and the positive regulation of apoptosis [45].

Our target analysis of this miRNA also revealed that *Pten* is a validated target in mice. Moreover, it was found that ~13-20% of miR-21a-5p targets are involved in the (A) negative regulation of ERK1 and ERK2 cascade—closely involved with cell proliferation—and (B) positive regulation of apoptosis. Note that both BP annotations included Pten, which was also a central node in the network analysis, demonstrating the importance of miR-21a-5p/Pten axis on the regulation of colon proliferation/apoptosis. In accordance with the downregulation of miR-21a-5p, CAF+CGA was the only treatment displaying a reduced Ki-67 labeling indexes on colonocytes and increased the percentage of colonocytes in apoptosis. Cell proliferation is closely related to (pre)neoplastic lesion development, as its increase may promote a clonal expansion of DMH-initiated epithelial cells, and ultimately promote ACF development [46,47] Furthermore, the ability to induce apoptosis in DMH-initiated epithelial cells may also prevent the emergence of (pre)neoplastic lesions [42]. As such, the modulation of crypt proliferation/apoptosis by coffee compounds may be involved with miR-21a-5p decrease, resulting in a decreased ACF burden.

DCA is a naturally occurring secondary bile acid that presents potential pro-carcinogenic and pro-inflammatory actions [14,48]. Some animal studies have demonstrated that mice receiving a DCA-supplemented diet developed gut dysbiosis and intestinal inflammation [48,49]. Our findings indicate that CAF+CGA intervention reduced the expression of pro-inflammatory cytokines IL-6, IL-17 and TNF-α in the colon. Note that Smad7, a negative regulator of pro-inflammatory TGF-β signaling, is a target of miR-21a-5p. When active, TGF-β signaling drives the pro-inflammatory shift of many stromal cells involved in the (pre)neoplastic lesion microenvironment [50]. The connection between inflammation and colon carcinogenesis is well stablished. The infiltration of CD8+ and CD3+ cells and other immune cell subsets in CRC have been associated with clinical prognoses and outcomes [51,52]. IL-6 is overexpressed in CRC patients and is correlated with a larger tumor size, the occurrence of liver metastasis and reduced survival rates, as this interleukin is also a potent stimulator of colon cancer cell proliferation [53,54,55]. The TNF-α is expressed initially in the first steps of inflammation, this cytokine is responsible for triggering many reactions, including the production of other cytokines, chemokines and endothelial adhesion molecules, besides increasing vascular permeability and recruiting immune cells to the site of infection [56,57,58,59]. Finally, the IL-17 is also expressed significantly higher in CRC tissues, and its upregulation begins in the adenoma stage and is at a higher level in the malignant stage. IL-17 promotes tumorigenesis through the production of myeloid-derived suppressor cells (MDSCs) and stimulates IL-6 secretion from stromal tumor cells activating the STAT3 pathway [59]. TNF- α and IL-17 presented a synergistic effect on the proliferation of HT-29 cells through stimulating the extracellular receptors of Kinase (ERK1/2) and increasing IL-17 downstream genes, such as MMP-9, MMP-7 and MMP-2 [60]. As such, it was suggested that the negative regulation of colonic mucosa inflammation by coffee compounds may be involved with a decrease in miR-21a-5p, also contributing to the decreased ACF burden.

## 4. Conclusions

The findings from our preclinical study indicate that the association of the most common bioactive compounds found in coffee beverages (CAF+CGA) attenuate early-stage colon carcinogenesis in a chemically induced model. These beneficial effects are probably mediated by the downregulation of an important oncomiR, thus modulating proliferation, apoptosis and inflammation. Our findings provide insights into which coffee compounds are involved on the recent reported protective effects of coffee intake on CRC outcomes in humans [26,29], and may inspire future clinical studies.

## 5. Materials and Methods

### 5.1. Experimental Design

Seven-week-old male Swiss Webster mice were randomly distributed into five experimental groups (*n* = 10/group) and submitted to a colon carcinogenesis model. In brief, mice were initiated for colon carcinogenesis by receiving one intraperitoneal (i.p.) injection of DMH per week [40 mg/kg body weight (b.wt.) in EDTA 0.0001 M; Merck KGaA, Darmstadt, Germany] for two weeks (weeks 1 and 2) [61]. To promote colon carcinogenesis after DMH initiation, the animals received a balanced diet supplemented with deoxycholic acid (DCA) at 0.02% (Merck KGaA, Darmstadt, Germany) for 10 weeks (weeks 3 to 13). Control mice received a DMH vehicle and non-supplemented balanced diet. Concomitantly to DCA intervention, mice received CAF (50 mg/kg b.wt./day), CGA (25 mg/kg b.wt./day), CAF+CGA (50 and 25 mg/kg b.wt./day, respectively) or just distilled water as a vehicle (intragastrically, five times a week) (8–10 am) for 10 weeks (Figure 5). The bioactive coffee compounds were diluted daily in distilled water. All mice were euthanized by exsanguination under ketamine/xylazine anesthesia (100/16 mg/kg b.wt. i.p.) at week 13. At necropsy, the liver was removed, weighted in order to evaluate absolute (g) relative (%) liver weights. The large intestine was removed and opened longitudinally and after a rapid macroscopic analysis, proximal, medial and distal parts were fixed in 10% formalin solution during 24 h for posterior histological and immunohistochemistry assays. A sample of the distal part (2 cm) was aseptically processed for molecular analysis, snapped frozen in liquid nitrogen, and stored at −80 °C.

All animals were obtained from ANILAB—Laboratory of Animals, Paulínia, São Paulo State, Brazil. Mice were kept in a room with continuous ventilation (16–18 air changes/h), relative humidity (45–65%), controlled temperature (20–24 °C) and a light/dark cycle of 12:12 h and were given water and a balanced diet (Nuvital, Quimtia, Colombo, Brazil) ad libitum. Body weight and food consumption were recorded once a week during the whole experiment. These protocols were approved by the Botucatu Medical School/UNESP Ethics Committee on the Use of Animals (CEUA) (Protocol number 1254/2017) and all animals received human care according to the criteria outlined in the “Guide for the Care and Use of Laboratory Animals” [62].

### 5.2. Dose Determination of Bioactive Coffee Compounds

High coffee consumption (~3 cups/day) leads to a ~13% to 37% lower risk of gastrointestinal cancer in humans [63,64]. Three cups/day is equivalent to a CAF intake of 200–300 mg/day (~2.8–4.0 mg/kg b.wt./day, considering a 70 kg person). The CAF (50 mg/kg b.wt./day) dose was calculated based on the allometric translation of the Human Equivalent Dose (HED) [65]. CGA dose (25 mg/kg b.wt.) was based in the CAF+CGA proportion in filtered coffee beverages (1:2). The intragastric administration (i.g.) of coffee compounds in mice, from seven weeks old (sexual maturity), simulates the human exposure to the coffee/CAF, where consumption begins from puberty through to the adult age [66]. The selected doses did not show toxic effects in rodents [22].

### 5.3. ACF Topographic Identification and Quantification

For ACF screening development (n = 10 animals/group), a classical preneoplastic colonic lesion [30] was made, the colon was removed, opened longitudinally, washed with distilled water, and measured (length, in cm). Samples of colon were fixed flat in 10% phosphate-buffered formalin solution for 24 h, then stored in ethanol 70%. Each colon sample was stained with 2% of methylene blue for 2 min, placed onto histological slides and then observed under conventional bright-field microscopy (Axiostar Plus, Zeiss, Oberkochen, Germany) at 20× magnification. For an ACF analysis, samples of proximal, medial and distal colonic mucosa were evaluated using well-stablished criteria [30]. Total number of ACF (mean per mice/group) and aberrant crypt (AC) (mean per mice/group) were calculated for each group. Moreover, considering those ACF that were ≥2 AC, the mean number of ACF or AC per colon length analyzed was calculated.5.4. Colonic Proliferation and Apoptosis Indexes

After ACF screening, colon samples were swiss-rolled and embedded in paraffin, sections were obtained and stained with haematoxylin-eosin (HE).

Immunostaining for Ki-67 (ab16667, 1:100, Abcam, UK) and cleaved caspase-3 (ab179817, 1:200, Abcam, Cambridge, UK) were conducted using specific primary antibodies. Five-micrometer sections of colon samples were deparaffinized and hydrated through xylene-alcohol-water graded series. Slides were submitted to antigen retrieval in 0.1 M citrate buffer in a pressure chamber (Dako Cytomation, Glostrup, Denmark), incubated with 3% hydrogen peroxide solution (10 min), and treated with skimmed milk (1 h). Slides were incubated with primary antibodies 4 °C overnight. The sections were washed with phosphate buffer saline (PBS) solution and incubated with biotinylated universal polymer (Erviegas, Indaiatuba, SP, Brazil) for 20 min. In order to stain the immunocomplexes, slides were incubated with 3′3-diaminobenzidine (DAB) chromogen solution (Sigma-Aldrich, St. Louis, MO, USA). Finally, the sections were counterstained with Harris’ hematoxylin. Immunostained sections were evaluated in conventional light microscopy (Olympus BX53, Tokyo, Japan). The proliferation and apoptosis indexes (PI% and AI%) were calculated in 20 randomly selected colonic crypts by dividing the number of Ki-67 or caspase-3 positive cells per total number of cells analyzed (*n* = 10 animals/group) [67].

### 5.4. Enzyme-Linked Imunosorbent Assay (ELISA)

Around 100 mg of colon samples (*n* = 7 animals/group) was homogenized in RIPA buffer (Cell Signaling, Danvers, MA, USA) containing 1% protease inhibitor cocktail (Sigma-Aldrich, USA), the proportion was 100 µL buffer for each 30 mg tissue, then maintained at 4 °C for 2 h. The homogenate content (tissue/buffer) was centrifuged (10,000× *g*, 4 °C, 30 min) and the supernatant was collected for the protein quantification using the Bradford method. Levels of tumor necrosis alpha (TNF-α), interleukin-17 (IL-17) and -6 (IL-6) were obtained by the Luminex multiple analyte profiling (xMAP) methodology using a 96-well plate with specific magnetic beads for each cytokine, based on the manufacturer’s instructions (MCYTOMAG-70 K, Millipore, Burlington, MA, USA).

### 5.5. RNA Isolation and miRNA Profiling

About 30 mg of colon samples (*n* = 5 animals/selected groups) were homogenized in 1 mL of QIAzol (Qiagen, Manchester, UK). Total RNA was isolated in each one using a QIAGEN RNeasy column-based system according to the manufacturer’s instructions (Qiagen, Hilden, Germany). The RNA quantification was assessed in a Qubit 2.0 fluorometer and its integrity was observed in Agilent 2100 Bioanalyzer platform (Agilent Technologies, Santa Clara, USA). Only samples with RNA integrity number (RIN) ≥ 7.0 were considered for the miRNA expression. The RNA samples were stored until the analysis at −80 °C.

A total of 100 ng of whole RNA was utilized for nCounter Mouse v1.5 miRNA global expression assay (efficient in detection of 600 murine and murine-related viral miRNAs, Appendix A) in an automated system (NanoString Technologies, Seattle, WA, USA). These analyses were carried out at the Molecular Oncology Research Center in the Barretos Cancer Hospital (Barretos, Brazil). The NanoString methodology consists in the incubation of total RNA samples with specific probes that bind 3′ ends of each mature miRNA normalizing the miRNA melting temperature. The tag excess was removed, the complexes that formed RNA-tags were incubated with 10 µL and 5 µL of reporter and capture probes, at 64 °C during 18 h. There were specific fluorescent signals for each complex in miRNA at the 5′end, and capture probes are biotinylated at the 3′end. Post purification, the mixture was pipetted in a streptavidin-covered cartridge by nCounter Prep Station. Finally, cartridges were analyzed in nCounter Digital Analyzer, acquiring 280 fields of view per sample and the miRNA-reporter probe complexes were counted. miRNA expression was analyzed by raw counting the miRNA-reporter probe complexes that had been normalized using the median of the top 100 miRNAs and that presented the lowest coefficient of variation (low CV values) using the NanoString package. A pair of comparisons were made considering *p* ≤ 0.05, a fold change (FC) > 1.5 and presented as log2 (FC). Commonly/differentially expressed miRNAs were evaluated using a Venn Diagram.

### 5.6. In Silico Analysis of miRNA Targets and Pathways

After the analysis of differentially modulated miRNAs, each miRNA was identified and classified on microT-CDS (v5.0), DIANA tools and mirPath v.3 (http://snf-515788.vm.okeanos.grnet.gr/; (accessed on 10 July 2019)). The output list of validated targets was submitted to the analysis of the functional enrichment of biological processes (BP) in the DAVID Bioinformatics Resource 6.8 online Platform (https://david.ncifcrf.gov/; (accessed on 10 July 2019)) [68]. Principal BP observations were organized by the high significance and lowest adjusted *p* values, considering *p* < 0.05. A network confidence analysis of the miRNA targets was carried out using the STRING database (https://string-db.org/; (accessed on 10 July 2019)).

### 5.7. Statistical Analysis

Data were analyzed using one-way ANOVA or Kruskal–Wallis and *post hoc* Tukey’s test and presented as mean ± standard deviation (SD). The number of samples per group for each analysis is represented by n. Statistical analysis was performed using GraphPad Prism Software 8.0 (GraphPad, San Diego, CA, USA), and differences were considered significant when *p* ≤ 0.05.

## Figures and Tables

**Figure 1 ijms-23-06292-f001:**
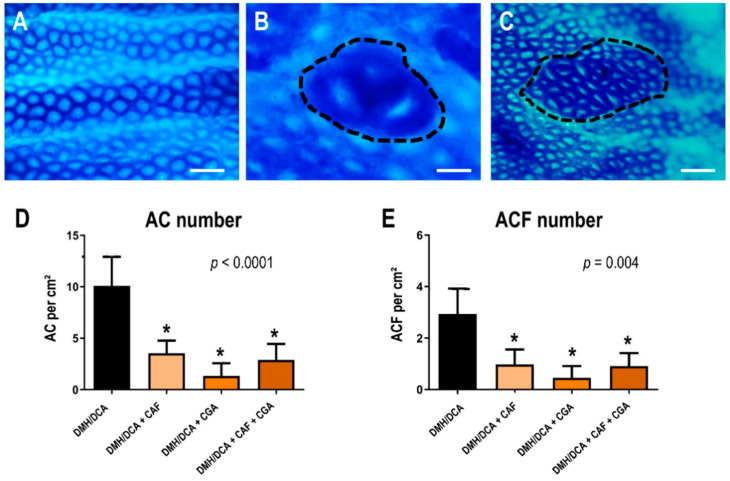
Representative methylene blue-stained colonic epithelium, showing (**A**) normal colon epithelium (scale bar: 100 µm, untreated group), (**B**) aberrant crypt foci with three crypts (scale bar: 50 µm, dotted line) and (**C**) aberrant crypt foci with >20 aberrant crypts (scale bar: 100 µm, dotted line). Effects of coffee compounds on the development of (**D**) AC or (**E**) ACF during DMH/DCA-induced colon carcinogenesis. *n* = 10 animals/group. Data are expressed as mean ± S.D. Untreated: 2×EDTA vehicle (i.p.)/balanced diet. DMH/DCA = 1,2-dimethylhydrazine (2 × 40 mg/kg b.wt., i.p.)/deoxycholic acid supplemented diet (0.02% *w*/*w*, 10 weeks). CAF = caffeine (50 mg/kg b.wt. intragastrical), CGA = chlorogenic acid (25 mg/kg b.wt. intragastrical) and CAF+CGA = caffeine + chlorogenic acid (50 and 25 mg/kg b.wt. intragastrical) for 10 weeks. * Statistical differences compared to the untreated group using ANOVA and Tukey’s test (*p* < 0.05).

**Figure 2 ijms-23-06292-f002:**
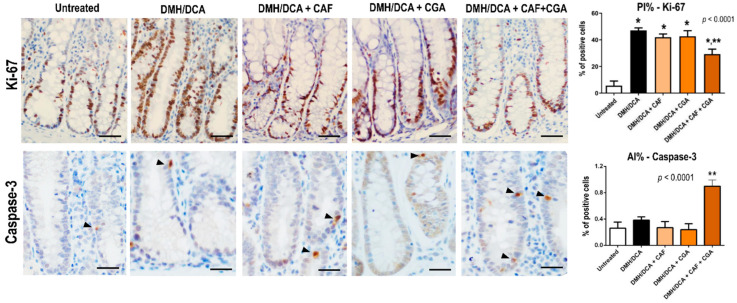
Effects of coffee compounds on colonic proliferation (PI%) and apoptosis indexes (AI%) during DMH/DCA-induced colon carcinogenesis. *n* = 10 animals/group. Data are expressed as mean ± S.D. Representative photomicrographs of Ki-67-positive colonocytes (scale bar: 50 µm) and cleaved caspase-3-positive apoptotic bodies (scale bar: 25 µm, arrowheads) are also displayed. Untreated: 2 × EDTA vehicle (i.p.)/balanced diet. DMH/DCA = 2 × 1,2-dimethylhydrazine (40 mg/kg b.wt., i.p.)/deoxycholic acid supplemented diet (0.02% *w*/*w*, 10 weeks). CAF = caffeine (50 mg/kg b.wt. intragastrical), CGA = chlorogenic acid (25 mg/kg b.wt. intragastrical) and CAF+CGA = caffeine + chlorogenic acid (50 and 25 mg/kg b.wt. intragastrical) for 10 weeks. Asterisks correspond to statistical differences compared to the untreated (*) or DMH/DCA (**) counterparts using ANOVA and Tukey test (*p* < 0.05).

**Figure 3 ijms-23-06292-f003:**
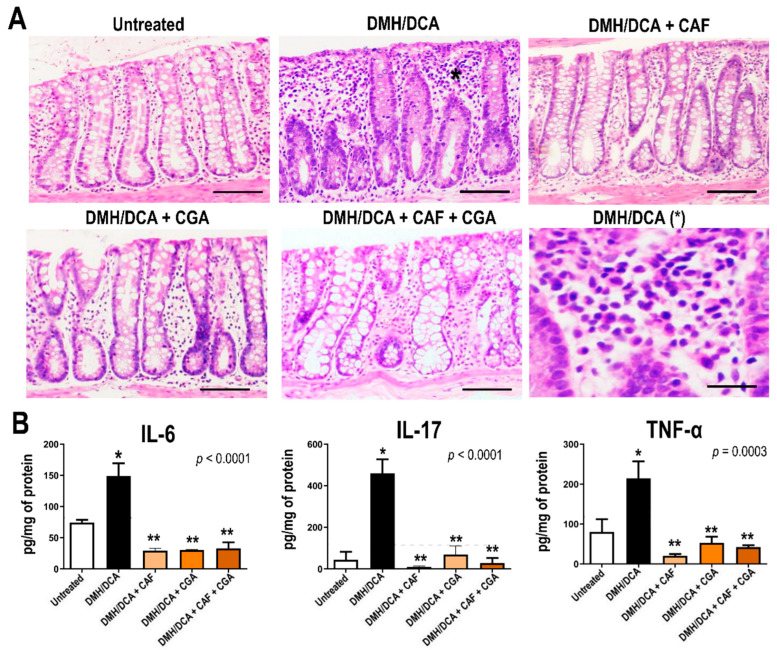
(**A**) Representative photomicrographs of HE-stained sections of colonic crypts (scale bar: 100 µm). * Detail on the inflammatory infiltrate present in the lamina propria of DMH/DCA group (scale bar: 20 µm). (**B**) Effects of coffee compounds on colonic levels of interleukins 6 (IL-6), 17 (IL-17) and the tumor necrosis factor alpha (TNF-α) during DMH/DCA-induced colon carcinogenesis. *n* = 7 animals/group. Data are expressed as mean ± S.D. Untreated: 2×EDTA vehicle (i.p.)/balanced diet. DMH/DCA = 2 × 1,2-dimethylhydrazine (40 mg/kg b.wt., i.p.)/deoxycholic acid supplemented diet (0.02% *w*/*w*, 10 weeks). CAF = caffeine (50 mg/kg b.wt. intragastrical), CGA = chlorogenic acid (25 mg/kg b.wt. intragastrical) and CAF+CGA = caffeine + chlorogenic acid (50 and 25 mg/kg b.wt. intragastrical) for 10 weeks. Asterisks correspond to statistical differences compared to the untreated (*) or DMH/DCA (**) counterparts using ANOVA and Tukey’s test (*p* < 0.05).

**Figure 4 ijms-23-06292-f004:**
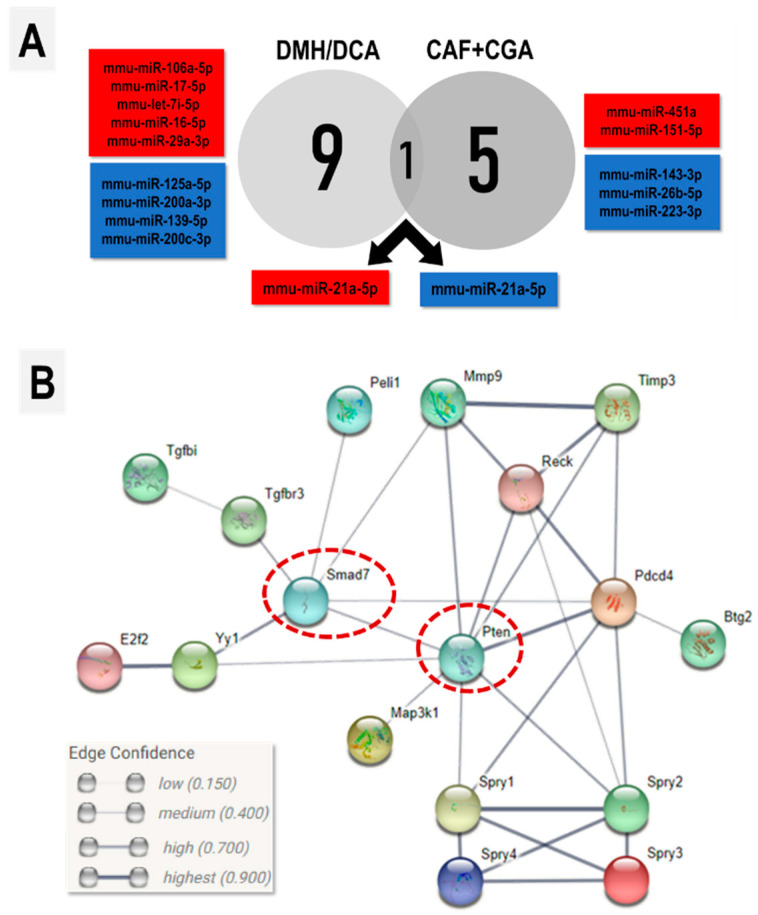
(**A**) The Venn diagram depicting the differentially expressed miRNAs (red: upregulated; blue: downregulated) in DMH/DCA (vs. untreated group) and CAF+CGA (vs. DMH/DCA). (**B**) STRING confidence network analysis of miR-21a-5p validated targets. Nodes in the correlated proteins are shown (with 3D structure inside). Edges correspond to the confidence of functional correlation (caption). DMH/DCA and CAF+CGA shared the differential expression of oncomiR miR-21-5p, which was upregulated in the DMH/DCA group and downregulated in CAF+CGA intervention. DMH/DCA = 2 × 1,2-dimethylhydrazine (40 mg/kg b.wt., i.p.)/deoxycholic acid supplemented diet (0.02% *w*/*w*, 10 weeks). CAF = caffeine (50 mg/kg b.wt. intragastrical), CGA = chlorogenic acid (25 mg/kg b.wt. intragastrical) and CAF+CGA = caffeine + chlorogenic acid (50 and 25 mg/kg b.wt. intragastrical) for 10 weeks.

**Figure 5 ijms-23-06292-f005:**
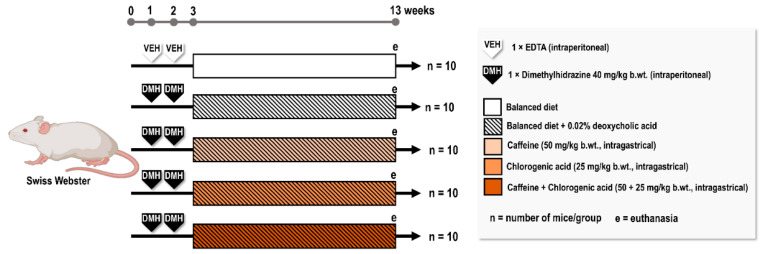
Experimental design.

**Table 1 ijms-23-06292-t001:** Effects of coffee compounds on food consumption, body weight, relative and absolute liver weights in DMH/DCA-induced colon carcinogenesis.

Groups/Treatments	Food Intake (g/mice/day)	Body Weight (g)			Absolute Liver Weight (g)	Relative Liver Weight (%)
		Initial	Final	Gain		
**Untreated**	6.19 ± 1.52	29.8 ± 2.5	40.3 ± 3.2	7.3 ± 3.6	2.0 ± 0.2	4.8 ± 0.4
**DMH/DCA**	5.69 ± 1.50	28.4 ± 1.6	37.1 ± 3.5	6.7 ± 4.2	2.4 ± 0.4	6.3 ± 1.0 *
**DMH/DCA+CAF**	5.57 ± 1.48	28.1 ± 2.8	36.8 ± 3.3	6.1 ± 4.4	2.8 ± 0.5	7.1 ± 1.5 *
**DMH/DCA+CGA**	5.72 ± 1.84	30.5 ± 2.8	37.0 ± 3.3	6.8 ± 4.3	2.5 ± 0.7	6.5 ± 0.8 *
**DMH/DCA+CAF+CGA**	5.61 ± 1.78	30.0 ± 1.6	38.4 ± 2.7	7. 7 ± 3.9	2.3 ± 0.3	6.8 ± 0.6 *

*n* = 10 animals/group. Data are expressed as mean ± S.D. Untreated: EDTA vehicle (i.p.)/balanced diet. DMH/DCA = 1,2-dimethylhydrazine (2 × 40 mg/kg b.wt., i.p.)/deoxycholic acid supplemented diet (0.02% *w*/*w*, 10 weeks). CAF = caffeine (50 mg/kg b.wt. intragastrical), CGA = chlorogenic acid (25 mg/kg b.wt. intragastrical) and CAF+CGA = caffeine + chlorogenic acid (50 and 25 mg/kg b.wt. intragastrical) for 10 weeks. * Statistical differences compared to the untreated group using ANOVA and Tukey’s test (*p* <0.05).

**Table 2 ijms-23-06292-t002:** miRNAs modulated in the DMH/DCA model and in CAF+CGA intervention group.

DMH/DCA vs. Untreated			DMH/DCA+CAF+CGA vs. DMH/DCA		
miRNA	Log2 (FC)	*p* Value	miRNA	Log2 (FC)	*p* Value
mmu-miR-21a-5p	2.32	0.03	mmu-miR-451a	2.09	0.001
mmu-miR-106a-5p	1.73	0.001	mmu-miR-151-5p	1.17	0.03
mmu-miR-17-5p	1.73	0.001	mmu-miR-21a-5p	−2.06	0.03
mmu-let-7i-5p	1.53	0.02	mmu-miR-143-3p	−1.58	0.04
mmu-miR-16-5p	1.42	0.009	mmu-miR-26b-5p	−1.51	0.04
mmu-miR-29a-3p	1.34	0.04	mmu-miR-223-3p	−1.45	0.02
mmu-miR-125a-5p	−1.65	0.01			
mmu-miR-200a-3p	−1.39	0.01			
mmu-miR-139-5p	−1.36	0.02			
mmu-miR-200c-3p	−1.29	0.01			

Considering Fold Change (FC) (>1.5) and *p* ≤ 0.05; Untreated: 2 × EDTA vehicle (i.p.)/balanced diet. DMH/DCA = 2 × 1,2-dimethylhydrazine (40 mg/kg b.wt., i.p.)/deoxycholic acid supplemented diet (0.02% *w*/*w*, 10 weeks). CAF = caffeine (50 mg/kg b.wt. intragastrical), CGA = chlorogenic acid (25 mg/kg b.wt. intragastrical) and CAF+CGA = caffeine + chlorogenic acid (50 and 25 mg/kg b.wt. intragastrical) for 10 weeks.

**Table 3 ijms-23-06292-t003:** Biological processes correlated to the validated targets of miR-21a-5p.

Biological Processes	Targets (% of Total)	Target Names	*p* Value
Negative regulation of ERK1 and ERK2 cascade	6 (18.8%)	Pten, Sprv1, Sprv2, Sprv3, Sprv4, Timp3	1.8 × 10^–7^
Negative regulation of epithelial to mesenchymal transition	4 (12.5%)	Pten, Sprv1, Sprv2, Tgfbr3	2.0 × 10^–5^
Regulation of signal transduction	4 (12.5%)	Sprv1, Sprv2, Sprv3, Sprv4	7.5 × 10^–5^
Negative regulation of TGF-β receptor signaling pathway	4 (12.5%)	Smad7, Sprv1, Sprv2, Tgfbr3	1.9 × 10^–4^
Positive regulation of apoptotic process	6 (18.8%)	Fasl, Mmp9, Map3k1, Moap1, Pten, Sprv1	2.6 × 10^–4^

## Data Availability

Data are available in this manuscript or from authors upon reasonable request.

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
