# Peer review of "Caffeine and Chlorogenic Acid Combination Attenuate Early-Stage Chemically Induced Colon Carcinogenesis in Mice: Involvement of oncomiR miR-21a-5p"

_ijms, 2022, doi:10.3390/ijms23116292_

Round 1

Reviewer 1 Report

Thank you for the opportunity to review the manuscript “Caffeine and chlorogenic acid combination attenuate early-2 stage chemically induced colon carcinogenesis in mice: involve-3 ment of oncomiR miR-21a-5p”. 
This is an interesting and well written study. It seems to be suitable for publication in its present form.  

Author Response

We thank the Reviewer for appreciating our work.

On behalf of all authors, I thank the reviewer’s comments on our manuscript. The authors highlight that the manuscript was submitted to an English revision in order to improve it (alterations in yellow).

Reviewer 2 Report

In this manuscript, Bartolomeu et al. evaluated the role of caffeine and chlorogenic acid in the attenuation of early-stage chemically induced colon carcinogenesis in vivo. The manuscript is written well and appropriately planned.

There are several issues that I would like the authors to tackle:

  1. The introduction is too long, it should briefly tackle the problem of colon cancer.
  2. The studies should also be accompanied by in vitro experiments using relevant cell lines.
  3. Could these results be transferred to other types of colon cancer? Not only induced by chemical compounds?

Author Response

We thank the Reviewer for appreciating our work.

The introduction is too long, it should briefly tackle the problem of colon cancer.

Author’s reply: The introduction was shortened, as suggested. The removed parts are highlighted as tracked changes. The authors highlight that the manuscript was also submitted to an English revision in order to improve it (alterations in yellow).

“The studies should also be accompanied by in vitro experiments using relevant cell lines.”

“Could these results be transferred to other types of colon cancer? Not only induced by chemical compounds?”

Author’s reply: The authors understand the reviewer’s concern, but these are issues are going to be explored in further investigations of our group. Actually, we are currently working on the effects of caffeine and/or chlorogenic acid on HT-29 and HCT116 spheroids.  Moreover, in order to validate our findings in another animal bioassay, HCT116 xenografts in NOD SCID mouse will be established.

Reviewer 3 Report

The article  describes  the protective effects of caffein and chlorogenic acid in a colon carcinogenesis mouse model. The article is well written and results are interesting. However, I have several concerns and some new experiments may give more information and improve this research.

1.- A H&E staining of different groups (including healthy mice) may be informative about the structure of the colonic epithelium and treatment effects. 

2.- Images of Ki67 stainig of the different groups and not only of one group should be added. Quantification is useful but the image of KI67 staining of untreated and the different treated animals is useful and informative.

3.- Authors claims CAF and CGA effects on apoptosis but they only analyze apoptotic bodies with a H&E staining. There are more specific and better techniques to analyze apoptosis. A TUNEL assay or another apoptosis marker should be added.

4.- The study of miRNAs modulated is interesing. Authors observed a downregulation of miR-21a-5p with CAF and CGA treatment. The target analysis of this miRNA shows several cancer-related proteins such as PTEN or Smad 7. Are these proteins regulated in animals treated with CAF or DGA?. This point would be very informative and should be clarified.

Author Response

We thank the Reviewer for appreciating our work.

“1.- A H&E staining of different groups (including healthy mice) may be informative about the structure of the colonic epithelium and treatment effects.”

Author’s reply: As suggested, the revised Fig.3A has representative photomicrographs of HE-stained section of all groups.

“2.- Images of Ki67 stainig of the different groups and not only of one group should be added. Quantification is useful but the image of KI67 staining of untreated and the different treated animals is useful and informative.”

Author’s reply: As suggested, the revised Fig.2 has representative photomicrographs of Ki-67 immunostanining of all groups.

“3.- Authors claims CAF and CGA effects on apoptosis but they only analyze apoptotic bodies with a H&E staining. There are more specific and better techniques to analyze apoptosis. A TUNEL assay or another apoptosis marker should be added.”

Author’s reply: As suggested, we performed immunohistochemistry for caspase-3. Results were similar to apoptotic bodies counting in HE, and can be seen in revised Fig. 2.  

“4.- The study of miRNAs modulated is interesing. Authors observed a downregulation of miR-21a-5p with CAF and CGA treatment. The target analysis of this miRNA shows several cancer-related proteins such as PTEN or Smad 7. Are these proteins regulated in animals treated with CAF or DGA?. This point would be very informative and should be clarified.”

Author’s reply: Since the most prominent results regarding proliferation and apoptosis, which are hallmarks directly related to miR-21a-5p, were observed in CAF+CGA treatment, we chose on performing miRNA global expression and target analysis only in this group.

Round 2

Reviewer 3 Report

The manuscript has been improved and the new version is suitable for publication. However there are a minor point that must be fixed. In caspase 3 graphic statistical significance is not clearly explained. In legend authors say:

Asterisks correspond to 177 statistical differences compared to the untreated (*) or DMH/DCA (**) counterparts using ANOVA 178 and Tukey test (P<0.05). However, in the Fig authors show two asterisks but p value indicates less than 0.0001 (this p value corresponds to 3 asterisks). There is no concordance between this information, which is the correct p value? How many asterisks are the correctly? This point should be clarify

Author Response

We thank the Reviewer for appreciating our work.

Author’s reply: The one-way ANOVA was performed, the p value refers to the ANOVA, and not to the adjusted p value specific to Tukey test comparison . In our paper, the asterisks are not relative to the value of p. The authors acknowledge that this is a common way to show the differences, but in all figures and tables (including Ki-67 data in the same figure as caspase-3) one asterisk refers to a significant difference compared to control group, while two refer to a significant difference compared to DMH, regardless of p value.